# SIMPLE YET EFFECTIVE SPATIO-TEMPORAL PROMPT LEARNING

## ABSTRACT

Accurate spatio-temporal prediction is pivotal for optimizing transportation systems and enhancing urban management. However, the practical application of cutting-edge graph neural network (GNN)-based methods for these tasks encounters challenges, particularly regarding their ability to generalize. GNN-based approaches have shown promise in capturing intricate spatial and temporal dependencies found in traffic and crime data. They utilize graph structures to model relationships between locations or entities, enabling the prediction of traffic patterns and crime incidents. Nonetheless, a key challenge involves ensuring that these models can effectively generalize to unseen scenarios and adapt to varying spatio-temporal data distributions. To tackle this challenge, we present a lightweight and effective prompt learning paradigm called as PromptST. This framework serves as an adaptation of pretrained spatio-temporal prediction models, specifically designed to handle the dynamics of spatial and temporal distributions. In the context of spatio-temporal prediction, our prompt tuning incorporates a simple prompt network into the pretrained model. By automatically learning informative prompt contexts that encapsulate the underlying spatial and temporal patterns from unseen data, the spatio-temporal prompt network guides the pretrained model to successfully adapt and learn from new data distributions. Our proposed prompt learning framework has been extensively evaluated on various spatio-temporal datasets, and the results demonstrate its effectiveness. Across multiple spatio-temporal prediction tasks, our PromptST achieves state-of-the-art prediction accuracy while maintaining computational efficiency, showcasing its superiority in capturing complex dependencies and adapting to varying data distributions across time and space.

## 1 INTRODUCTION

Spatio-temporal prediction in urban computing is a vital task that involves forecasting and estimating future states, events, or patterns within urban environments, considering both spatial and temporal dimensions. It has greatly enhanced the modeling of intricate dependencies and improved prediction performance in a wide range of domains, such as traffic prediction (Lan et al., 2022), crime forecasting (Li et al., 2022), and environmental monitoring (Yi et al., 2018). Recently, significant strides have been made in the field of spatio-temporal graph neural networks (GNNs) to achieve state-of-the-art performance in spatio-temporal prediction (Jin et al., 2023b; Wang et al., 2022). Spatio-temporal GNNs leverage the power of graph structures by extending convolutional and attentive neural networks. They excel at capturing spatial relationships and effectively propagating information across the graph, enabling them to model intricate dependencies present in spatio-temporal urban data (Shao et al., 2022). The learning process of these models involves training on spatio-temporal data to learn the model parameters, which are then used to make predictions on unseen testing data.

Significant progress has been made in the development of spatio-temporal graph neural architectures. However, a fundamental challenge arises from the assumption of a consistent distribution between the training and testing spatio-temporal data. In real-life urban scenarios, the testing distribution often experiences uncontrolled and unknown shifts compared to the training distribution. These shifts can be attributed to changes in urban dynamics, evolving environmental conditions, or shifts in population behavior. For instance, traffic patterns may be influenced by infrastructure developments or alterations in commuting habits. Additionally, unpredictable events like large-scale gatherings or urban crimes can have a substantial impact on spatio-temporal patterns. As a result, the presence

of these distribution shifts poses a significant obstacle to the accurate performance of existing spatio-temporal prediction models, leading to inaccurate predictions and reduced reliability.

In this study, we present a new spatio-temporal prompt learning paradigm called PromptST, designed to improve the generalization capability of spatio-temporal prediction models when dealing with unseen data and changing distributions. Our approach draws inspiration from the successful application of prompt-tuning techniques in the field of textual data (Wei et al., 2021; Yao et al., 2022). Specifically, by integrating a specially designed prompt neural network into pretrained models, PromptST enables effective adaptation and generalization to the unseen spatio-temporal prediction context. Our paradigm allows downstream tasks to provide customized prompts that explicitly guide the predictions of sophisticated yet computationally intensive spatio-temporal neural networks.

Our lightweight PromptST framework brings two key benefits for advancing spatio-temporal prediction: i) **Efficiency**. It allows for fine-tuning specific prompt parameters rather than the entire model, resulting in faster training and inference time. By updating only a subset of parameters in our simple spatio-temporal prompt network, we effectively mitigate the risk of overfitting that can occur with heavy, pretrained spatio-temporal neural architectures. ii) **Adaptability**. Our PromptST provides a flexible mechanism to adapt the spatio-temporal model's behavior to distribution shift. As our prompt learning component is specifically designed to capture the relevant spatio-temporal context and distribution characteristics from unseen data, it allows the model to align its predictions with the shifted data distribution. In a nutshell, this work makes the following main contributions:

- This work aims to enhance the adaptability of spatio-temporal models in effectively addressing distribution shifts, which are frequently encountered in real-world scenarios characterized by spatial and temporal dynamics. The improved adaptability of the model enables it to maintain high performance even when faced with changing or previously unseen data.

- This study presents PromptST, a novel spatio-temporal prompt learning paradigm that integrates a simple prompt network with prediction models, allowing for effective adaptation and generalization in customized spatio-temporal contexts. Additionally, in-depth analysis is provided to justify the model's capability in alleviating distribution shifts and achieving enhanced efficiency.

- We perform extensive experiments on a range of spatio-temporal prediction tasks using diverse datasets to thoroughly evaluate the effectiveness, efficiency, and robustness of our proposed framework. Through comparisons with numerous state-of-the-art methods, we obtain compelling results that strongly validate the superiority of our approach. To facilitate result reproducibility, we provide the model implementation which can be accessed at https://anonymous.4open.science/r/PromptST.

## 2 RELATED WORK

**Spatio-Temporal Forecasting.** Spatio-temporal prediction holds significant importance in a wide range of domains, such as transportation, epidemiology, and public safety. To tackle this challenge, considerable efforts have been devoted to developing various neural network architectures capable of capturing complex spatial and temporal correlations across different time slots and locations. For instance, recurrent neural networks (RNNs) (Lv et al., 2018; Ding et al., 2022), attention mechanisms (Luo et al., 2021; Xu et al., 2020) and temporal convolutional networks (Wu et al.; Bai et al., 2020), have been successfully employed to capture the transitional patterns present in spatio-temporal data, further enhancing the predictive capabilities of the models. Recent advancements have focused on designing spatio-temporal graph neural networks (GNNs) (Zhu et al.; Han & Gong, 2022; Lan et al., 2022) to effectively encode spatial correlations among different locations. These approaches leverage the inherent graph structure of the data, allowing for the high-order modeling of interactions between spatially connected nodes, thereby enhancing the accuracy and provide state-of-the-art performance. However, a prevailing challenge faced by most existing spatio-temporal models is their limited capability to handle distribution shifts in the context of spatio-temporal dynamics.

**Prompt-tuning Techniques**. The objective of prompt-tuning is to optimize and fine-tune prompts in order to improve the performance and generalization of pretrained models on specific tasks or domains. Inspired by the success of prompt-tuning in language modeling, researchers have extended its application to various domains, such as computer vision (Jia et al., 2022; Bahng et al., 2022) and graph neural networks (GNNs) (Sun et al., 2022; Fang et al., 2022). For example, Liu etal Fang et al. (2022) unifies graph pre-training and downstream tasks into a common task template. Motivated

Figure 1: Architecture Overview of the Spatio-Temporal Prompt Learning Paradigm PromptST.

by these advancements, this work introduces a spatio-temporal prompt learning framework that leverages the success of prompt-tuning to customize and refine pretrained models, providing an effective solution to improve their adaptability in the context of spatio-temporal dynamics.

## 3 METHODOLOGY

This section provides a detailed elaboration on the proposed PromptST framework, including the spatio-temporal prediction task, the spatio-temporal prompt tuning paradigm, and in-depth theoretical analysis on our PromptST framework. Figure 1 illustrates the overall architecture of PromptST.

### 3.1 SPATIO-TEMPORAL PREDICTION

Spatio-temporal (ST) prediction focuses on forecasting data that is spatially and temporally distributed in urban scenarios. Examples of such predictions include estimating traffic volumes on roads (Pan et al., 2019) and predicting the number of crime cases in different regions (Huang et al., 2018).

**Spaio-Temporal Data**. Formally, the target spatio-temporal data can be denoted by a three-way tensor $\mathcal{X} \in \mathbb{R}^{R \times T \times F}$, where $R$ denotes the number of spatial regions (*e.g.* urban districts, road segments), $T$ denotes the number of time slots (*e.g.* quarters, hours, days), and $F$ denotes the dimensionality of the concerned features (*e.g.* the number of crime types). An element $\mathcal{X}_{r,t,f} \in \mathbb{R}$ denotes the value of the $f$-th feature for the $r$-th region in the $t$-th time slot. And $\mathcal{X}_t \in \mathbb{R}^{R \times F}$ is the time-specific matrix.

**Spaio-Temporal Graph**. Besides the spatio-temporal data $\mathcal{X}$, it is common to work with a spatio-temporal graph that records the correlations between the regions and the time slots. This graph can be represented as $\mathcal{G} = (\mathcal{V}, \mathcal{E}, \mathbf{X})$, where $\mathcal{V}$ is a set of nodes representing urban regions, and $\mathcal{E}$ is the set of edges that encode both the spatial interrelations and the temporal transition relations between the region nodes. The node set $\mathcal{V}$ is associated with a node feature matrix $\mathbf{X} = \{\mathbf{x}_1, \mathbf{x}_2, ..., \mathbf{x}_{|\mathcal{V}|}\} \in \mathbb{R}^{|\mathcal{V}| \times d}$, where $\mathbf{x}_i \in \mathbb{R}^d$ represents the $d$-dimensional feature vector of the node $v_i$.

**Task Formalization**. Based on the above definitions, the spatio-temporal prediction task is to predict the future values of the ST tensor $\mathcal{X}$, with the help of the historical records of $\mathcal{X}$ and the ST graph $\mathcal{G}$. This can be formalized as learning a spatio-temporal model $g(\cdot)$ with parameter set $\boldsymbol{\Theta}_g$ as follows:

$$(\mathcal{X}_{t+1}, \mathcal{X}_{t+2}, \cdots, \mathcal{X}_{t+T'}) = g(\mathcal{X}_{t-T+1}, \mathcal{X}_{t-T+2}, \cdots, \mathcal{X}_t, \mathcal{G}; \boldsymbol{\Theta}_g) \tag{1}$$

### 3.2 SPATIO-TEMPORAL PROMPT TUNING

#### 3.2.1 PRETRAINING AND TUNING PARADIGM

In real urban scenarios, spatio-temporal data typically exhibits daily variations, resulting in a dynamic and evolving distribution. However, these variations pose a challenge for the existing ST models trained using static method, limiting their ability to accurately predict new data that is temporally distant from the historical training samples. To fill this gap, our PromptST framework adopts the pretraining and tuning paradigm. It requires the ST model, pretrained on fixed historical data, to continuously adapt to newly updated data. This enables the model to maintain its predictive accuracy over time, effectively capturing the evolving nature of the spatio-temporal data. In this pretraining and tuning paradigm, we split the entire ST data into three subsets $\mathcal{X} = (\mathcal{X}_{pre}, \mathcal{X}_{tun}, \mathcal{X}_{tst})$, as follows:

$$\mathcal{X}_{pre} = \mathcal{X}_{t-T+1:t-T+T_{pre}}, \quad \mathcal{X}_{tun} = \mathcal{X}_{t-T+T_{pre}+1:t}, \quad \mathcal{X}_{tst} = \mathcal{X}_{t+1:t+T'} \tag{2}$$

Here, the pretraining data $\mathcal{X}_{pre}$, tuning data $\mathcal{X}_{tun}$, and test data $\mathcal{X}_{tst}$ are arranged in chronological order. The index $t$ denotes the time slot separating the test data from the other two sets. $T$ indicates the

total number of time slots in the combined pretraining and tuning data, $T_{pre}, T'$ represents the length of the training data and the test data, respectively. Following the pretraining and tuning paradigm, ST models are pretrained on $\mathcal{X}_{pre}$, finetuned on the new data $\mathcal{X}_{tun}$, and ultimately evaluated on $\mathcal{X}_{tst}$.

A common approach following this paradigm is the pretrain-and-finetune method (Fang et al., 2022), which involves fine-tuning a pretrained model on newly updated data for model adaptation. However, the fine-tuning process can be computationally expensive, particularly for heavy but accurate ST models based on GNNs. In light of the effectiveness and efficiency of prompt tuning (Sun et al., 2023) in adapting pretrained models to unseen data, our PromptST proposes a prompt tuning approach for ST prediction, enabling efficient adaptation. Specifically, instead of fine-tuning the entire ST model on new data, PromptST fixes the pretrained model during the tuning process, and optimizes a prompt network for model adaption. This prompt tuning method can be described as the objective below:

$$\underset{\boldsymbol{\Theta}_h}{\arg\min} \ \mathcal{L}(g(h(\mathcal{X}_{tun}, \mathcal{G}; \boldsymbol{\Theta}_h); \boldsymbol{\Theta}_g), \mathcal{X}_{tun}), \ \text{where} \ \boldsymbol{\Theta}_g = \underset{\boldsymbol{\Theta}_g}{\arg\min} \mathcal{L}(g(\mathcal{X}_{pre}; \boldsymbol{\Theta}_g), \mathcal{X}_{pre}) \quad (3)$$

where $h(\cdot), g(\cdot)$ denote the prompt neural network and the original ST model, respectively. Their contained parameters are referred as $\boldsymbol{\Theta}_h, \boldsymbol{\Theta}_g$, respectively. $\mathcal{L}(\cdot)$ denotes the loss function such as the squared mean error. This equation presents the two-step optimization process of our prompt tuning framework. In the first phase, the original ST model $g(\cdot)$ is optimized using the pretraining data $\mathcal{X}_{pre}$. Here $g(\cdot)$ can be any proposed ST model. In the second phase, our PromptST optimizes the plug-in prompt neural network $h(\cdot)$ using the newly updated data $\mathcal{X}_{tun}$. It mitigates the distribution shift of the new data $\mathcal{X}_{tun}$ by learning a transformation, adapting $\mathcal{X}_{tun}$ to the pretrained ST model $g(\cdot)$ customized to the old data $\mathcal{X}_{pre}$. Compared to fine-tuning the entire ST model $f(\cdot)$, our prompt neural network $g(\cdot)$ is able to achieve comparable or even better performance with much less training steps, due to its lightweight network architecture which is easier to optimize.

### 3.2.2 Time-aware Prompt Network

To capture the essential temporal dependencies that impact the region relationships in spatio-temporal data, we integrate a temporal convolutional network (TCN) into our prompt neural network. This fusion introduces dynamism into the transformed features. In formal terms, the operation of our PromptST, which combines the prompt neural network with TCN, can be described as follows:

$$\tilde{\mathcal{X}}_{r,t} = \mathbf{W}_4\bar{\mathbf{H}}_{r,t} + \mathcal{X}_{r,t}, \ \bar{\mathbf{H}}_r = \sigma(\mathbf{W}_3\tilde{\mathbf{H}}_r + \mathbf{b}_2), \ \tilde{\mathbf{H}}_r = \sigma(\delta(\mathbf{W}_2 * \mathbf{H}_r + \mathbf{b}_1)), \ \mathbf{H}_{r,t} = \mathbf{W}_1\mathcal{X}_{r,t} \quad (4)$$

where $\tilde{\mathcal{X}} \in \mathbb{R}^{R \times T_{tun} \times F}$ denotes the spatio-temporal data transformed by our prompt network, with $T_{tun} = T - T_{pre}$ denoting the number of time slots in the tuning data. $\bar{\mathbf{H}} \in \mathbb{R}^{R \times T_{tun} \times d}$ denotes the intermediate embedding with hidden dimensionality $d$. The results of the TCN, which convolves the original $T_{tun}$ temporal dimensions into $T'_{tun}$ dimensions, are denoted by $\tilde{\mathbf{H}} \in \mathbb{R}^{R \times T'_{tun} \times d}$. $\mathbf{H} \in \mathbb{R}^{R \times T_{tun} \times d}$ denotes the initial embeddings for all regions and time slots. The learnable parameters of our prompt neural network are $\mathbf{W}_4 \in \mathbb{R}^{F \times d}$, $\mathbf{W}_3 \in \mathbb{R}^{T_{tun} \times T'_{tun}}$, $\mathbf{W}_2 \in \mathbb{R}^{(T_{tun} - T'_{tun} + 1) \times 1}$, $\mathbf{W}_1 \in \mathbb{R}^{d \times F}$, and $\mathbf{b}_1, \mathbf{b}_2 \in \mathbb{R}^d$. And $*$ denotes the convolution operator. $\sigma(\cdot), \delta(\cdot)$ denote the ReLU activation and the dropout function, respectively. A skip connection is utilized in the final layer of our prompt network, to directly utilize the original ST data. The output $\tilde{\mathcal{X}}$ of our prompt network has the same dimensionality as the original ST data $\mathcal{X}$, and thus can be seamlessly used by any ST model.

### 3.3 In-depth Discussion

This section makes further discussions to study two research questions: i) How does the prompt network alleviate the distribution shift of spatio-temporal data? ii) How is the efficiency of the proposed PromptST in comparison to the fine-tuning method and spatio-temporal prediction baselines?

**Prompt Network as Data Projector**. Our PromptST model leverages vanilla spatio-temporal prediction loss functions, such as mean absolute error (MAE), to optimize and maximize the accuracy of the final spatio-temporal (ST) prediction task. While no explicit constraints are placed on mitigating distribution shifts, we demonstrate that our prompt network design is inherently trained to function as a data editor for the original ST data. This is evident through the following observations:

$$\frac{\partial \mathcal{L}}{\partial \theta} = \frac{\partial \mathcal{L}}{\partial \tilde{\mathcal{X}}} \cdot \frac{\partial \tilde{\mathcal{X}}}{\partial \theta} = \frac{\partial \mathcal{L}}{\partial (\mathcal{X} + \nabla \mathcal{X})} \cdot \frac{\partial \nabla \mathcal{X}}{\partial \theta}, \ \text{where} \nabla \mathcal{X} = \mathbf{W}_4 \bar{\mathbf{H}} \quad (5)$$

Here $\nabla \mathcal{X}$ denotes the adjustable output of our prompt network. By incorporating a skip connection to include the original data $\mathcal{X}$, $\nabla \mathcal{X}$ functions as an editing component. The equation presented decomposes the gradient of the loss function $\mathcal{L}$ with respect to a specific learnable parameter $\theta$ from the prompt network into two components: the gradient of $\mathcal{L}$ with respect to the edited ST data, and the gradient of the editing value with respect to the parameter $\theta$. In essence, the training objective of our PromptST is to learn a spatio-temporal data editor that yields improved performance.

**Model Efficiency Analysis**. We demonstrate the efficiency advantages of our PromptST from two perspectives: a comparison of the number of parameters and a comparison of time complexity. Firstly, we observed that the optimization operations in both the pretraining phase and the tuning phase are nearly identical. The only difference lies in the parameter set used, as illustrated below:

$$\text{Pretraining: } \theta := \theta - \eta \cdot \frac{\partial \mathcal{L}}{\partial \theta}, \ \theta \in \mathbf{\Theta}_g; \quad \text{Prompt Tuning: } \theta := \theta - \eta \cdot \frac{\partial \mathcal{L}}{\partial \theta}, \ \theta \in \mathbf{\Theta}_h \tag{6}$$

The two phases employ the same MAE loss function $\mathcal{L}$ (Wu et al.) and the same learning rate $\eta$. However, there is a significant difference in the number of parameters, as empirically $|\mathbf{\Theta}_g| > C \times |\mathbf{\Theta}_h|$ where $|\mathbf{\Theta}|$ denotes the number of parameters and $C \geq 10$. This confers a substantial efficiency advantage to the tuning phase of our PromptST, not only by reducing the number of optimization operations but also by facilitating easier convergence during the optimization process.

Secondly, the pretrained model is GNN-based which has a higher time complexity due to its costly graph information propagation paradigm. While our lightweight prompt tuning network consists of only 2-layer TCN and 2 fully-connected layers. Specifically, the time complexity of the GNN-based pretrained model is $\mathcal{O}(|\mathcal{E}| \times L \times d)$, where $|\mathcal{E}|$ denotes the number of edges, and $L$ denotes the number of graph layers. In contrast, the prompt tuning neural network $f(\cdot)$ only requires $\mathcal{O}(L' \times d^2)$ for prompt training, where $L'$ represents the number of MLP layers, and $d$ is the hidden dimensionality.

In conclusion, our analysis of the number of parameters and time complexity shows that the proposed prompt learning model outperforms GNN-based ST models in terms of tuning efficiency. This makes it a promising framework for large-scale data in real-world spatial-temporal prediction scenarios.

# 4 EXPERIMENTS

We assess the performance of our PromptST through evaluations on two spatio-temporal prediction tasks: traffic prediction and crime forecasting. The experiments aim to address the following research questions: 1) **RQ1**: How does our PromptST compare to various state-of-the-art baselines in terms of performance? 2) **RQ2**: What is the impact of each component on the performance of our PromptST? 3) **RQ3**: How does the efficiency of our PromptST compare to other state-of-the-art methods? 4) **RQ4**: What effects do different hyperparameter settings have on the performance of our PromptST?

## 4.1 EXPERIMENTAL SETUP

**Datasets.** We evaluate PromptST's performance on traffic prediction and crime prediction using a set of real-life datasets. These datasets include 8 traffic flow datasets (PeMSD04, PeMSD07, PeMSD03, PeMSD08, and PeMS-Bay as point-based datasets, and NYTaxi, CHIBike, and TDrive as grid-based datasets) as well as 2 crime datasets from New York and Chicago. The traffic data is collected at varying intervals of 5 minutes, 30 minutes, or 60 minutes. Following previous studies (Bai et al., 2020; Diao et al., 2019), we build the urban spatial graph upon the road network for traffic datasets. For crime prediction, four crime types are employed. Specific data statistics are presented in Table 7.

**Evaluation Metrics.** We closely follow the same settings in (Li et al., 2018; Chen et al., 2011; Zhu et al.; Li & Zhu, 2021) by utilizing the same evaluation metrics and data splits. For point-based traffic datasets, we predict the traffic in the next 12 time slots using traffic records in the previous 12 time steps. For the grid-based traffic prediction, we adopt 2 historical time steps to predict the next 2 time steps. Three metrics are used for evaluating both tasks, including Mean Absolute Error (MAE), Mean Absolute Percentage Error (MAPE) and Root Mean Squared Error (MAPE). For crime prediction task, we follow the same setting of (Xia et al., 2022) and use MAE and MAPE as metrics.

**Baselines.** We conduct a comparative analysis of our PromptST framework against various baselines on two spatio-temporal prediction tasks. To ensure a fair comparison for traffic prediction, we adopt

Table 1: Overall performance of traffic prediction on PeMSD4, PeMSD8, PeMSD3 and PeMSD7.

| Models | PeMSD04 | | | PeMSD08 | | | PeMSD03 | | | PeMSD07 | | | PeMS-Bay | | |
|---|---|---|---|---|---|---|---|---|---|---|---|---|---|---|---|
| | MAE | RMSE | MPAE | MAE | RMSE | MPAE | MAE | RMSE | MAPE | MAE | RMSE | MAPE | MAE | RMSE | MAPE |
| HA | 38.03 | 59.24 | 27.88% | 34.86 | 52.04 | 24.07% | 31.58 | 52.39 | 33.78% | 45.12 | 65.64 | 24.51% | 2.88 | 5.59 | 6.82% |
| VAR | 24.54 | 38.61 | 17.24% | 19.19 | 29.80 | 13.10% | 23.65 | 38.26 | 24.51% | 50.22 | 75.63 | 32.22% | 2.32 | 5.25 | 5.61% |
| DSAN | 22.79 | 35.77 | 17.12% | 17.14 | 26.96 | 11.32% | 21.29 | 34.55 | 23.21% | 31.36 | 49.11 | 14.43% | 2.16 | 4.97 | 5.54% |
| DCRNN | 24.70 | 38.12 | 14.17% | 17.86 | 27.83 | 11.45% | 17.99 | 30.31 | 18.34% | 25.22 | 38.61 | 11.82% | 2.07 | 4.74 | 4.90% |
| STGCN | 22.70 | 35.55 | 14.59% | 18.02 | 27.83 | 11.40% | 17.55 | 30.42 | 17.34% | 25.33 | 39.34 | 11.21% | 2.42 | 5.33 | 5.58% |
| GWN | 25.45 | 39.70 | 17.29% | 19.13 | 31.05 | 12.68% | 19.12 | 32.77 | 18.89% | 26.39 | 41.50 | 11.97% | 1.95 | 4.52 | 4.63% |
| ASTGCN | 22.93 | 35.22 | 16.56% | 18.25 | 28.06 | 11.64% | 17.34 | 29.56 | 17.21% | 24.01 | 37.87 | 10.73% | 2.10 | 4.77 | 5.30% |
| LSGCN | 21.53 | 33.86 | 13.18% | 17.73 | 26.76 | 11.30% | 17.94 | 29.85 | 16.98% | 27.31 | 41.16 | 11.98% | 2.13 | 4.82 | 5.18% |
| STSGCN | 21.19 | 33.65 | 13.90% | 17.13 | 26.86 | 10.96% | 17.48 | 29.21 | 16.78% | 24.26 | 39.03 | 10.21% | 2.10 | 4.74 | 5.28% |
| AGCRN | 19.83 | 32.26 | 12.97% | 15.95 | 25.22 | 10.09% | 15.98 | 28.25 | 15.23% | 22.37 | 36.55 | 9.12% | 1.96 | 4.57 | 4.69% |
| STFGNN | 19.83 | 31.88 | 13.02% | 16.64 | 26.22 | 10.60% | 16.77 | 28.34 | 16.30% | 22.07 | 35.80 | 9.21% | 1.83 | 4.33 | 4.19% |
| STGODE | 20.84 | 32.82 | 13.77% | 16.81 | 25.97 | 10.62% | 16.50 | 27.84 | 16.69% | 22.99 | 37.54 | 10.14% | 2.02 | 4.40 | 4.72% |
| Z-GCNETs | 19.50 | 31.61 | 12.78% | 15.76 | 25.11 | 10.01% | 16.64 | 28.15 | 16.39% | 21.77 | 35.17 | 9.25% | 2.03 | 4.38 | 4.71% |
| TAMP | 19.74 | 31.74 | 13.22% | 16.36 | 25.98 | 10.15% | 16.46 | 28.44 | 15.37% | 21.84 | 35.42 | 9.24% | 2.04 | 4.45 | 4.76% |
| DSTAGNN | **19.30** | **31.46** | **12.70%** | 15.67 | 24.77 | 9.94% | 15.57 | 27.21 | 14.68% | 21.42 | 34.51 | 9.01% | 2.13 | 4.79 | 5.32% |
| FOGS | 19.74 | 31.66 | 13.05% | 15.73 | 24.92 | **9.88%** | 15.89 | 25.74 | 15.13% | 21.28 | 34.88 | 8.95% | 2.07 | 4.51 | 4.80% |
| *PromptST* | 19.42 | 31.54 | 13.02% | **15.52** | **24.69** | 10.06% | **14.87** | **24.73** | **14.29%** | **20.72** | **33.37** | **8.76%** | **1.70** | **3.81** | **3.77%** |

Table 2: Performance of PromptST based on different backbones on PeMS-Bay data.

| Mertics | ASTGCN | | | STGCN | | | MTGNN | | | AGCRN | | | STSGCN | | |
|---|---|---|---|---|---|---|---|---|---|---|---|---|---|---|---|
| | MAE | RMSE | MAPE | MAE | RMSE | MAPE | MAE | RMSE | MAPE | MAE | RMSE | MAPE | MAE | RMSE | MAPE |
| Pretrained Model | 2.28 | 4.53 | 5.44% | 2.43 | 5.47 | 6.01% | 2.01 | 4.32 | 4.65% | 1.84 | 4.02 | 4.22% | 2.21 | 4.96 | 5.26% |
| Prompt Tuning | 1.96 | 4.31 | 4.64% | 2.03 | 4.45 | 4.88% | 1.70 | 3.81 | 3.77% | 1.72 | 3.79 | 3.83% | 1.87 | 4.10 | 4.21% |

the same experimental settings as described in the studies (Chen et al., 2021a; Rao et al., 2022) for predicting point-based traffic data (*i.e.*, PeMSD03, PeMSD04, PeMSD07, PeMSD08, PeMS-Bay) and the research works (Yao et al., 2018; Yao et al.) for forecasting grid-based traffic data (*i.e.*, NYCTaxi, T-Drive, CHIBike). For crime prediction, we select the baselines based on the same experimental setup as presented in (Xia et al., 2021). Baseline details can be found in Appendix A.7.

## 4.2 OVERALL EFFECTIVENESS

We evaluate the effectiveness of our PromptST by comparing it to baselines on traffic prediction and crime prediction. PromptST in these experiments utilizes the lightweight MTGNN as the backbone model. We also evaluate the performance of PromptST with different backbone models. The results are presented in Table 1 (point-based traffic prediction), Table 8 (grid-based traffic prediction), Table 3 (crime prediction), and Table 2 (impact of backbone models). We make the following observations.

- **Superior performance**: Our PromptST framework consistently outperforms the state-of-the-art baseline methods across the three evaluated tasks: grid-based traffic prediction, point-based traffic prediction, and crime prediction. This demonstrates the effectiveness of our prompt learning network in capturing distribution shifts between the pretrained and tuning data. In contrast, existing baselines experience a decline in performance due to the distribution gap. Furthermore, the notable performance gain observed in the crime prediction task showcases the capability of PromptST in handling sparse and heterogeneous spatio-temporal data, such as crimes. These advantages can be primarily attributed to our carefully designed spatio-temporal prompt tuning paradigm with the successful injection of spatio-temporal context distilled from the downstream data.

- **Significant improvements on large-scale data**: It is important to highlight that our PromptST exhibits a considerably larger performance gap compared to the baselines when evaluated on the large-scale dataset PeMS-Bay. This outcome can be attributed to the heightened difficulty faced by non-adaptive baselines in handling the substantial distribution shift present in PeMS-Bay, which encompasses a broader time range. In contrast, our PromptST effectively addresses this challenge by adeptly adapting itself to the shifted domain through its prompt tuning network.

- **Variations among different backbone models**: While our prompt tuning paradigm in PromptST effectively mitigates the distribution shift issue for all backbone models listed in Table 2, a notable difference in performance is observed, particularly for STGCN and STSGCN compared to the other three backbones. We attribute the larger performance gap for STGCN and STSGCN to their relatively weaker generalization ability, as they solely rely on GNNs for spatio-temporal modeling, making them more susceptible to overfitting the observed data. In contrast, MTGNN and AGCRN incorporate TCN modules to capture temporal dynamics, while ASTGCN incorporates an auxiliary temporal modeling view using the GRU network. Such diverse modeling techniques enhance the generalization ability of these models, resulting in a smaller disparity in performance.

Table 3: Overall oerformance of urban crime prediction on NYC and CHI datasets.

| Model | New York City | | | | | | | | Chicago | | | | | | | |
|---|---|---|---|---|---|---|---|---|---|---|---|---|---|---|---|---|
| | Burglary | | Larceny | | Robbery | | Assault | | Theft | | Battery | | Assault | | Damage | |
| | MAE | MAPE | MAE | MAPE | MAE | MAPE | MAE | MAPE | MAE | MAPE | MAE | MAPE | MAE | MAPE | MAE | MAPE |
| ARIMA | 0.8999 | 0.6305 | 1.3015 | 0.6268 | 0.9558 | 0.5969 | 0.9983 | 0.6198 | 1.5965 | 0.5720 | 1.3212 | 0.5792 | 0.8691 | 0.6044 | 1.0430 | 0.6134 |
| SVM | 1.1604 | 0.7653 | 1.4979 | 0.6417 | 1.1278 | 0.6733 | 1.1928 | 0.6964 | 1.7711 | 0.5629 | 1.3493 | 0.6027 | 1.0879 | 0.6560 | 1.1313 | 0.5721 |
| STResNet | 0.8680 | 0.5603 | 1.1082 | 0.5329 | 0.8717 | 0.5209 | 0.9645 | 0.5749 | 1.3931 | 0.5488 | 1.1519 | 0.5719 | 0.7679 | 0.4633 | 0.9064 | 0.5018 |
| DCRNN | 0.8176 | 0.5324 | 1.0732 | 0.5492 | 0.9189 | 0.5532 | 0.9692 | 0.5955 | 1.3699 | 0.5770 | 1.1583 | 0.5528 | 0.7639 | 0.4600 | 0.8764 | 0.4756 |
| STGCN | 0.8366 | 0.5404 | 1.0629 | 0.5295 | 0.9035 | 0.5441 | 0.9375 | 0.5757 | 1.3628 | 0.5359 | 1.1512 | 0.5761 | 0.7963 | 0.4810 | 0.9068 | 0.4959 |
| GWN | 0.7993 | 0.5235 | 1.0493 | 0.5405 | 0.8681 | **0.5351** | 0.8866 | 0.5646 | 1.3211 | 0.5502 | 1.1331 | 0.5503 | 0.7493 | 0.4580 | 0.8584 | 0.4850 |
| STtrans | 0.8617 | 0.5592 | 1.0896 | 0.5478 | 0.8839 | 0.5651 | 0.9363 | 0.5679 | 1.3404 | 0.5356 | 1.1466 | 0.5684 | 0.7671 | 0.4499 | 0.8987 | 0.4842 |
| DeepCrime | 0.8227 | 0.5508 | 1.0618 | 0.5351 | 0.8841 | 0.5537 | 0.9222 | 0.5677 | 1.3391 | 0.5430 | 1.1290 | 0.5389 | 0.7737 | 0.4616 | 0.9096 | 0.4960 |
| STDN | 0.8831 | 0.5768 | 1.1442 | 0.5889 | 0.9230 | 0.5649 | 0.9498 | 0.5661 | 1.5303 | 0.6287 | 1.2076 | 0.5791 | 0.8052 | 0.4820 | 0.9169 | 0.4869 |
| ST-MetaNet | 0.8285 | 0.5369 | 1.0697 | 0.5627 | 0.9214 | 0.5766 | 0.9323 | 0.5702 | 1.3369 | 0.5369 | 1.1762 | 0.5748 | 0.7904 | 0.4753 | 0.8907 | 0.4756 |
| GMAN | 0.8652 | 0.5633 | 1.0503 | 0.5340 | 0.9234 | 0.5671 | 0.9338 | 0.5803 | 1.3235 | 0.5307 | 1.1442 | 0.5560 | 0.7852 | 0.4714 | 0.8823 | 0.4838 |
| AGCRN | 0.8260 | 0.5397 | 1.0499 | 0.5404 | 0.9013 | 0.5383 | 0.9063 | **0.5519** | 1.3281 | 0.5304 | 1.1432 | 0.5697 | 0.7669 | 0.4612 | 0.8712 | 0.4859 |
| STSHN | 0.8012 | 0.5198 | 1.0431 | **0.5291** | 0.8717 | 0.5362 | 0.9169 | 0.5682 | 1.3231 | 0.5310 | 1.1348 | **0.5544** | 0.7758 | 0.4574 | 0.8741 | **0.4747** |
| DMSTGCN | 0.8376 | 0.5485 | 1.0410 | 0.5464 | 0.8597 | 0.5403 | 0.9036 | 0.5601 | 1.3292 | **0.5291** | **1.1297** | 0.5552 | 0.8058 | 0.4759 | 0.8698 | 0.4877 |
| *PromptST* | **0.7117** | **0.4874** | **1.0404** | 0.5689 | **0.8105** | 0.5506 | **0.9005** | 0.6119 | **1.2854** | 0.5457 | 1.1531 | 0.6161 | **0.7049** | **0.4273** | **0.8398** | 0.4964 |

Table 4: Tuning time (minutes) comparison with different amount of tuning data for traffic prediction.

| Datasets | PeMSD04 | | | PeMSD07 | | | PeMSD03 | | | PeMSD08 | | |
|---|---|---|---|---|---|---|---|---|---|---|---|---|
| Time line | 1 day | 1 week | 2 weeks | 1 day | 1 week | 2 weeks | 1 day | 1 week | 2 weeks | 1 day | 1 week | 2 weeks |
| Time for Training Scratch | 2.382 | 13.913 | 20.401 | 9.093 | 30.905 | 59.940 | 10.560 | 22.130 | 27.716 | 1.071 | 3.477 | 7.152 |
| Time for Finetune | 2.848 | 14.620 | 23.036 | 8.002 | 40.865 | 20.541 | 10.227 | 20.754 | 16.381 | 2.025 | 2.945 | 8.432 |
| Time for Prompt Tuning | 1.302 | 1.952 | 5.604 | 5.694 | 12.220 | 13.959 | 3.030 | 3.071 | 10.782 | 0.489 | 1.371 | 3.103 |
| Faster x than Scratch | 45.340% | 85.970% | 72.531% | 37.380% | 60.460% | 76.712% | 71.307% | 86.123% | 61.098% | 54.342% | 60.570% | 56.614% |
| Faster x than Prompt | 54.284% | 86.648% | 75.673% | 50.948% | 70.097% | 32.043% | 70.373% | 85.203% | 34.180% | 75.852% | 53.447% | 63.200% |
| Datasets | NYCTaxi | | | T-Drive | | | CHIBike | | | PeMS-Bay | | |
| Time line | 1 day | 1 week | 2 weeks | 1 day | 1 week | 2 weeks | 1 day | 1 week | 2 weeks | 1 week | 2 weeks | 4 weeks |
| Time for Training Scratch | 1.477 | 21.181 | 45.808 | 1.341 | 36.651 | 43.290 | 1.232 | 12.130 | 19.002 | 17.328 | 32.187 | 40.326 |
| Time for Finetune | 1.125 | 10.304 | 13.170 | 1.015 | 14.843 | 15.726 | 1.076 | 8.355 | 10.137 | 9.284 | 25.193 | 31.728 |
| Time for Prompt Tuning | 0.967 | 7.683 | 10.102 | 0.902 | 10.137 | 11.784 | 0.895 | 6.044 | 8.128 | 6.777 | 21.506 | 27.748 |
| Faster x than Scratch | 34.529% | 63.727% | 77.947% | 32.737% | 72.342% | 72.780% | 27.354% | 50.173% | 57.226% | 60.890% | 33.184% | 31.190% |
| Faster x than Prompt | 14.044% | 25.437% | 23.295% | 11.133% | 31.705% | 25.067% | 16.822% | 27.660% | 19.818% | 27.003% | 14.635% | 12.544% |

## 4.3 EFFICIENCY COMPARISON

We study the efficiency of our PromptST framework by comparing it to two tuning techniques: training randomly-initialized ST models on the tuning data, and fine-tuning pretrained ST models on the tuning data. The tuning time from start to model convergence is recorded. The evaluation is done on a server with 10 cores of Inter(R) Core(TM) i9-9820X CPU@3.30GHZ, 64GB RAM, and 4 NVIDIA Geforce RTX 3090 GPUs. In Table 4, we present the running time of tuning models with 1-day, 1-week, and 2-week tunning data. MTGNN is employed as the backbone in this experiment. In Table 5 and Table 9, we show the tuning time of using different backbone models, using two-week and one-week tuning data, respectively. From the results we have the following major conclusions.

**1) Efficiency of prompt tuning**: The advantageous tuning efficiency of our PromptST is demonstrated by its less tuning time on different datasets. PromptST only needs to tune the smaller parameter set of the prompt network. This not only reduces the computational overhead for optimization calculations but also facilitates easier convergence by constraining the solution space of the model. **2) Comparing fine-tuning to training from scratch**: While both compared tuning techniques optimize the same number of model parameters, we generally observe higher tuning efficiency with the fine-tuning method. This can be attributed to the pretraining process, which provides better starting points for the fine-tuning method, enabling faster convergence. However, there are instances where training from scratch outperforms fine-tuning. These cases reflect the uncertainty of whether the pretrained model state is advantageous or detrimental for training on the tuning data. **3) Efficiency under different tuning data size**: As the amount of tuning data increases (1 day, 1 week, and 2 weeks), the tuning time for all three methods also increases due to the growing complexity of the data. The efficiency of our PromptST is further confirmed by its ability to maintain and even enhance its efficiency advantage when dealing with larger tuning sets. **4) Impact of backbone models**: By referring to Table 5 and Table 9, it becomes evident that PromptST effectively expedites the tuning process for various ST models, irrespective of their size. This attribute holds significant value in real-world applications.

## 4.4 ABLATION STUDY

To assess the effectiveness of each component in our PromptST framework, we conduct ablation experiments on both tasks. The evaluation results for traffic prediction and crime prediction can be found in Table 6 and Figure 2, respectively. We examine the following ablated variants: **1) w/o TCN**: This variant removes the temporal convolutional network from our prompt neural network. It exhibits

Table 5: Tuning time (minutes) comparison for different backbones with two-week tuning data.

| Datasets | PeMSD04 | | | | | PeMSD07 | | | | |
|---|---|---|---|---|---|---|---|---|---|---|
| Models | ASTGCN | STGCN | MTGNN | AGCRN | STSGCN | ASTGCN | STGCN | MTGNN | AGCRN | STSGCN |
| Time for Training Scratch | 118.733 | 24.765 | 20.401 | 24.966 | 44.755 | 334.768 | 83.992 | 59.940 | 75.284 | 145.894 |
| Time for Finetune | 92.796 | 22.176 | 23.036 | 19.012 | 39.785 | 280.456 | 76.501 | 20.541 | 32.785 | 126.733 |
| Time for Prompt Tuning | 74.535 | 17.864 | 9.379 | 10.347 | 27.667 | 256.114 | 53.121 | 15.959 | 11.068 | 89.756 |
| Faster x than Scratch | 37.225% | 27.866% | 54.027% | 58.556% | 38.181% | 23.495% | 36.755% | 52.920% | 85.293% | 38.479% |
| Faster x than Prompt | 19.679% | 19.444% | 59.286% | 45.576% | 30.459% | 8.679% | 30.562% | 22.307% | 66.241% | 29.177% |

Table 6: Ablation study for the proposed PromptST on the large-scale traffic data PeMS-Bay.

| Datasets | PeMS-Bay (1 week) | | | PeMS-Bay (2 weeks) | | | PeMS-Bay (3 weeks) | | | PeMS-Bay (4 weeks) | | |
|---|---|---|---|---|---|---|---|---|---|---|---|---|
| Metrics | MAE | RMSE | MAPE | MAE | RMSE | MAPE | MAE | RMSE | MAPE | MAE | RMSE | MAPE |
| PromptST | 1.63 | 3.68 | 3.63% | 1.65 | 3.71 | 3.67% | 1.68 | 3.90 | 3.79% | 1.70 | 3.81 | 3.77% |
| w/o TCN | 1.65 | 3.70 | 3.66% | 1.67 | 3.73 | 3.70% | 1.71 | 3.94 | 3.83% | 1.74 | 3.92 | 3.85% |
| w/o MLP | 1.69 | 3.72 | 3.69% | 1.70 | 3.76 | 3.72% | 1.75 | 3.96 | 3.87% | 1.80 | 4.12 | 3.94% |
| w/o data initial | 1.70 | 3.74 | 3.71% | 1.72 | 3.78 | 3.75% | 1.78 | 3.99 | 3.91% | 1.86 | 4.11 | 3.97% |

significant performance decay in certain types of crime data (e.g., Burglary), highlighting the importance of explicit modeling of temporal relationships in certain spatio-temporal prediction scenarios for our PromptST. **2) w/o MLP**: In this variant, we eliminate the multiple fully-connected layers in the prompt network. Notable performance degradation is observed in most cases, underscoring the necessity of employing transformation layers to adapt to the distribution shift of the newly generated data. **3) w/o Skip**: This ablated model eliminates the skip connection in the final layer of our prompt neural network. Without this design, the prompt network faces challenges in generating appropriate input spatio-temporal data for the pretrained backbone model $g(\cdot)$. The evaluation results confirm this difficulty, as this version experiences a significant performance drop in most cases.

In addition to evaluating the effectiveness of the components in our PromptST, Table 6 further confirms that tuning data covering a larger temporal range generally results in a larger distribution shift, leading to more pronounced performance degradation for the pretrained model. This observation reinforces the motivation behind our PromptST, which aims to develop an effective prompt tuning approach to adapt to temporal shifts successfully. Through comparing the ablated versions with the full version of our PromptST, it can be concluded that the inclusion of all components in PromptST enhances its robustness against the increase in distribution disparity.

## 4.5 HYPERPARAMETER STUDY

In this section, we investigate the influence of different hyperparameter settings on prediction accuracy and tuning time. The evaluation is carried out on both traffic flow prediction using the PeMSD04 and PeMSD08 datasets, as well as crime prediction using data from New York City and Chicago. The results, in terms of MAE, are presented in Figure 3, while results for other metrics are shown in Figure 5. Specifically, we examine the following hyperparameters:

- **Embedding dimensionality**: The optimal performance for our PromptST model is achieved with an embedding dimensionality of 32 for both tasks. When comparing the impact on performance between the two tasks, we observe that the embedding dimensionality has a more significant effect on the crime prediction task. This can be attributed to the periodic and noise-free characteristics of traffic data, which mitigate the risks of severe under-fitting and over-fitting, respectively. In terms of efficiency, a larger embedding dimensionality leads to a substantial increase in tuning time. A model with a dimensionality of 32 strikes a balance, requiring a moderate amount of tuning time.

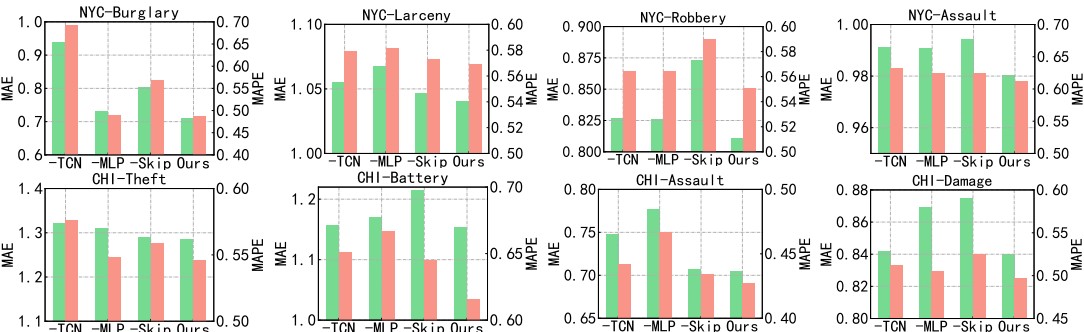

Figure 2: Ablation study of PromptST on crime prediction

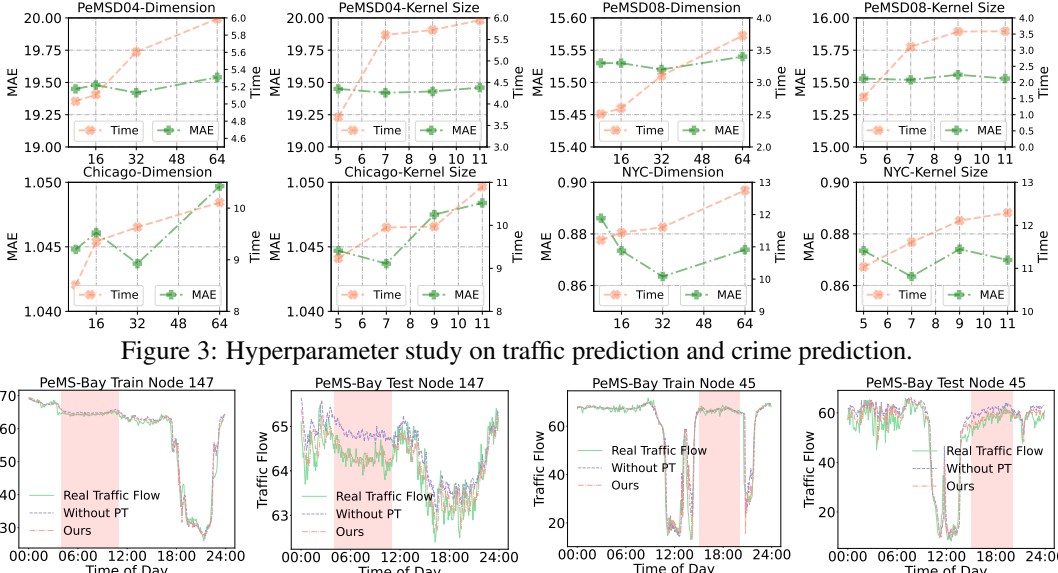

Figure 3: Hyperparameter study on traffic prediction and crime prediction.

Figure 4: Case study of PromptST on PeMS-Bay to show data distribution shift

- **Kernel size of TCN**: This parameter determines the number of consecutive time slots considered in the temporal relation modeling of our prompt network. Based on the results, a kernel size of 7 demonstrates performance advantages in certain cases. Similar to the embedding dimensionality, a more pronounced impact is observed for the crime prediction task. Additionally, a kernel size of 7 proves to be highly efficient in terms of tuning time within our PromptST framework.

## 4.6 CASE STUDY

In this section, we assess the effectiveness of the proposed PromptST framework in mitigating spatio-temporal distribution shifts by examining specific cases. Figure 4 illustrates the variation in traffic flow throughout the day for two region nodes: 147 and 45. The left plot for each region represents the training data, while the right plot represents the test data. Each plot includes three curves: the ground truth traffic flow, the predicted values obtained using the ablated model without prompt tuning (referred to as "Without PT"), and the predicted values obtained using the full version of our PromptST. We summarize the key observations as follows:

In the training data, both our PromptST and the ablated model without prompt tuning exhibit high prediction accuracy compared to the ground truth curves. However, the ground truth traffic data from the test set for the same regions demonstrates a noticeable distribution shift. Specifically, the curve in the red region for node 147 shows significant oscillations, while the curve in the red region for node 45 exhibits a distinct rising trend that differs from the training data. In comparison to our PromptST, the predictions made by the ablated model without prompt tuning display clear inaccuracies. This validates the strong capability of PromptST in addressing distribution shifts effectively.

## 5 CONCLUSION

In this study, we introduce a simple yet powerful spatio-temporal prompt learning paradigm aimed at enhancing the robustness and generalization ability of spatio-temporal prediction models in the presence of dynamic distribution shifts. Our framework incorporates prompt tuning, which involves generating informative spatio-temporal prompt context that captures the underlying patterns and dynamics in the downstream urban data. Through comprehensive empirical evaluations across various spatio-temporal prediction tasks, we have demonstrated the remarkable effectiveness of our spatio-temporal prompt learning framework. By leveraging this framework, we significantly improve the resilience of pre-trained models to distribution shifts and enhance their adaptability to new data.

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

# A   APPENDIX

## A.1   ALGORITHM

---

**Algorithm 1:** The PromptST Learning Algorithm

---

**Input:** The spatial-temporal graph $\mathcal{G}$, the maximum epoch number $E$, the learning rate $\eta$;
**Output:** Traffic flow or crime rate $\mathbf{H}$ and trained parameters in $\Theta_f$ of Prompt neural network
         and $\Theta_g$ of GNN-based neural network;

1   Initialize all parameters in $\Theta_g$ and $\Theta_f$;
2   Train the framework PromptST by Equation 3
3   **for** $epoch = 1, 2, ..., E$ **do**
4      Split the date into train, test and prompt;
5      Train the GNN-based pretrain model via train dataset;
6      **for** $\theta_g \in \Theta_g$ **do**
7         $\theta_g = \theta_g - \eta \cdot \frac{\partial \mathcal{L}}{\partial \theta_g}$
8      **end**
9   **end**
10   **for** $epoch = 1, 2, ..., E$ **do**
11      Freeze parameters of GNN-baed pretrain model and update the prompt neural network via
         Equation 3 via prompt dataset;
12      Compute the MAE loss $\mathcal{L}$ following Equation 3;
13      Minimize the loss $\mathcal{L}$ by Equation 6 using gradient decent with learning rate $\eta$;
14      **for** $\theta_f \in \Theta_f$ **do**
15         $\theta_f = \theta_f - \eta \cdot \frac{\partial \mathcal{L}}{\partial \theta_f}$
16      **end**
17   **end**
18   **Return** $\mathbf{H}$ and all parameters $\Theta_g$ and $\Theta_f$;

---

The Algorithm 1 section of our framework PromptST presents specific algorithmic specifics. The initialization of all the parameters is the first step, as seen in Algorithm 1. After then, the GNN-based model is trained until it is proficient via updating parameters of $\Theta_g$. We train the prompt tuning neural network iteratively and fix the GNN-based model. To improve the performance of the prompt tuning neural network, we employ the MAE loss Wu et al. in accordance with earlier studies that are mentioned in the traffic prediction task. With this approach, the MAE loss is determined after E 3. We tune the prompt tuning neural network 6 until it converges. Following all these steps, the procedure ends and returns all $\Theta_g$ and $\Theta_f$ parameters.

## A.2   EVALUATION METRICS AND EVALUATION PLATFORM

Following existing studies of traffic flow prediction Bai et al. (2020); Li & Zhu (2021); Fang et al. (2021); Chen et al. (2021a;b); Rao et al. (2022); Lan et al. (2022), we adopt three widely utilized metrics namely Mean Absolute Error (MAE), Mean Absolute Percentage Error (MAPE) and Root Mean Squared Error (RMSE) as evaluation metrics for traffic prediction of 5 point-based datasets, namely PeMSD04, PeMSD08, PeMSD03, PeMSD07 and PeMSD-Bay in Table 1, and 3 grid-based datasets, namely NYCTaxi, T-Drive and CHIBike. For crime prediction task, we follow the settings in Xia et al. (2022) in terms of Mean Absolute Error (MAE) and Mean Absolute Percentage Error (MAPE) metrics on NYC crime and Chicago crime datasets. All methods are implemented in Python 3.9 and PyTorch 1.12.0. The experiments are conducted on a server with 10-cores of Intel(R) Core(TM) i9-9820X CPU @ 3.30GHz 64.0GB RAM and 4 Nvidia GeForce RTX 3090 GPU.

## A.3   EFFECTIVENESS

We conducted experiments on grid-based datasets, specifically NYCTaxi, T-Drive, and CHIBike, to evaluate the performance of our model, PromptST, in terms of inflow and outflow predictions. The results are summarized in Table 8. Upon analysis, we observe that our model consistently achieves the best performance across most cases and demonstrates superior performance in the remaining

Table 7: Data Description of 10 Datasets

| Traffci Data | Point-based Datasets | | | | | Grid-based Datasets | | |
|---|---|---|---|---|---|---|---|---|
| Datasets | PeMSD04 | PeMSD08 | PeMSD03 | PeMSD07 | PeMS-Bay | NYTaxi | CHIBike | TDrive |
| Sensors | 307 | 170 | 358 | 883 | 325 | 75 (15 × 5) | 270 (15 × 18) | 1024 (32 × 32) |
| Data | 16,992 | 17,856 | 26,208 | 28,224 | 52,116 | 17,520 | 4,416 | 3,600 |
| Interval | 5 minutes | 5 minutes | 5 minutes | 5 minutes | 5 minutes | 30 minutes | 30 minutes | 60 minutes |
| Crime Data | NYC-Crimes | | | | | Chicago-Crimes | | |
| Time Span | Jan, 2014 to Dec, 2015 | | | | | Jan, 2016 to Dec, 2017 | | |
| Category | Burglary | | | Robbery | | Theft | | Battery |
| Cases | 31,779 | | | 33,453 | | 124,630 | | 99,389 |
| Categoty | Assult | | | Larceny | | Damage | | Assult |
| Cases | 40,429 | | | 85,899 | | 59,886 | | 37,972 |

Table 8: Overall performance of Grid-based Datasets of Traffic Prediction

| Datasets | NYCTaxi | | | | | | T-Drive | | | | | | CHIBike | | | | | |
|---|---|---|---|---|---|---|---|---|---|---|---|---|---|---|---|---|---|---|
| Metrics | inflow | | | outflow | | | inflow | | | outflow | | | inflow | | | outflow | | |
| Models | MAE | MAPE | RMSE | MAE | MAPE | RMSE | MAE | MAPE | RMSE | MAE | MAPE | RMSE | MAE | MAPE | RMSE | MAE | MAPE | RMSE |
| STResNet | 14.492 | 14.543 | 24.050 | 12.798 | 14.368 | 20.633 | 19.636 | 17.831 | 34.890 | 19.616 | 18.502 | 34.597 | 4.767 | 31.382 | 6.703 | 4.627 | 30.571 | 6.559 |
| DMVSTNet | 14.377 | 14.314 | 23.734 | 12.566 | 14.318 | 20.409 | 19.599 | 17.683 | 34.478 | 19.531 | 17.621 | 34.303 | 4.687 | 32.113 | 6.635 | 4.594 | 31.313 | 6.455 |
| DSAN | 14.287 | 14.208 | 23.585 | 12.462 | 14.272 | 20.294 | 19.384 | 17.465 | 34.314 | 19.290 | 17.379 | 34.267 | 4.612 | 31.621 | 6.695 | 4.495 | 31.256 | 6.367 |
| DCRNN | 14.421 | 14.353 | 23.876 | 12.828 | 14.344 | 20.067 | 22.121 | 17.750 | 38.654 | 21.755 | 17.382 | 38.168 | 4.236 | 31.264 | 5.992 | 4.211 | 30.822 | 5.824 |
| STGCN | 14.377 | 14.217 | 23.860 | 12.547 | 14.095 | 19.962 | 21.373 | 17.539 | 38.052 | 20.913 | 16.984 | 37.619 | 4.212 | 31.224 | 5.954 | 4.148 | 30.782 | 5.779 |
| GWN | 14.310 | 14.198 | 23.799 | 12.282 | 13.685 | 19.616 | 19.556 | 17.187 | 36.159 | 19.550 | 15.933 | 36.198 | 4.151 | 31.153 | 5.917 | 4.101 | 30.690 | 5.694 |
| STSGCN | 15.604 | 15.203 | 26.191 | 13.233 | 14.698 | 21.653 | 23.825 | 18.547 | 41.188 | 24.287 | 19.041 | 42.255 | 4.256 | 32.991 | 5.941 | 4.265 | 32.612 | 5.879 |
| STFGNN | 15.336 | 14.869 | 26.112 | 13.178 | 14.584 | 21.627 | 22.144 | 18.094 | 40.071 | 22.876 | 18.987 | 41.037 | 4.234 | 32.222 | 5.933 | 4.264 | 32.321 | 5.875 |
| STGODE | 14.621 | 14.793 | 25.444 | 12.834 | 14.398 | 20.205 | 21.515 | 17.579 | 38.215 | 22.703 | 18.509 | 40.282 | 4.169 | 31.165 | 5.921 | 4.125 | 30.726 | 5.698 |
| STGNCDE | 14.281 | 14.171 | 23.742 | 12.276 | 13.681 | 19.608 | 19.347 | 17.134 | 36.093 | 19.230 | 15.873 | 36.143 | 4.123 | 31.151 | 5.913 | 4.094 | 30.595 | 5.678 |
| STTN | 14.359 | 14.206 | 23.841 | 12.373 | 13.762 | 19.827 | 20.583 | 17.327 | 37.220 | 20.443 | 15.992 | 37.067 | 4.160 | 31.208 | 5.932 | 4.118 | 30.704 | 5.723 |
| GMAN | 14.267 | 14.114 | 23.728 | 12.273 | 13.672 | 19.594 | 19.244 | 17.110 | 35.986 | 18.964 | 15.788 | 36.120 | 4.115 | 31.150 | 5.910 | 4.090 | 30.662 | 5.675 |
| TFormer | 13.995 | 13.912 | 23.487 | 12.211 | 13.611 | 19.522 | 18.823 | 16.910 | 34.470 | 18.883 | 15.674 | 35.219 | 4.071 | 31.141 | 5.878 | 4.037 | 30.647 | 5.638 |
| ASTGNN | **13.844** | **13.692** | **23.177** | 12.112 | 13.602 | **19.201** | 18.798 | 16.101 | 33.870 | 18.790 | 15.584 | 33.998 | 4.068 | 31.131 | **5.818** | 3.981 | 30.617 | 5.609 |
| PromptST | 14.123 | 13.762 | 23.569 | **12.103** | **13.316** | 19.462 | **18.173** | **15.456** | **32.417** | **18.342** | **15.407** | **32.293** | **4.021** | **31.103** | 5.875 | **3.745** | **29.017** | **5.398** |

cases as well. We attribute this success to the following factors: (1) The integration of a prompt tuning neural network, which incorporates Temporal Convolutional Networks (TCN), proves beneficial in capturing temporal features. This ability to capture and leverage temporal information plays a crucial role in accurately predicting traffic flows. (2) Our model utilizes a residual paradigm, where the initial data is added to the model. This approach ensures that our model maintains the same data distribution as the input of the pre-trained model. This helps to preserve the integrity of the data and contributes to the improved performance of our model. By leveraging these strategies, our model PromptST demonstrates superior performance in traffic flow predictions. The incorporation of the prompt tuning neural network and the residual paradigm effectively capture temporal features and maintain data distribution, respectively, resulting in enhanced prediction accuracy.

## A.4 HYPERPARAMETER STUDY

We conducted a hyperparameter study on four datasets: PeMSD04, PeMSD08, Chicago and NYC crime datasets. The study aimed to investigate the impact of two hyperparameters on model performance: the dimension, ranging from 16 to 128, and the kernel size, ranging from 5 to 11. The evaluation metric used was Mean Absolute Percentage Error (MAPE), and the results are illustrated in Figure 5. Upon analysis, we observed that our model achieved the best performance when the dimension was set to 32 and the kernel size was set to 7. It is worth noting that setting a larger dimension may lead to oversmoothing in the GNN-based backbone model, which can subsequently degrade the performance of the prompt neural network. On the other hand, a larger kernel size may introduce more noise from the traffic data, ultimately reducing the overall performance. By carefully selecting the hyperparameters, we are able to optimize the performance of our model. The findings provide valuable insights for achieving better results in traffic flow predictions.

## A.5 HYPERPARAMETER SETTINGS

For fair comparison, all compared algorithms have hidden dimensionality modified from the range [8,16,32,64] to achieve their best performance as reported results at 32. The learning rate $\eta$ is initialized as 0.003 with weight decay 0.3. For GNN-based models, the number of GCN layer is 3. For prompt tuning network, the number of the TCN Layer is 2 and the number of MLP layer is set as 2. The kernel size of the TCN Layer is set as 7 during which our framework PromptST obtains the best performance from the range of [5,7,9,11]. Following existing settings of traffic prediction, we

Table 9: Comparison of Time of Different Methods (One Week) (Minutes)

| Datasets | PeMSD04 | | | | | PeMSD07 | | | | |
|---|---|---|---|---|---|---|---|---|---|---|
| Models | ASTGCN | STGCN | MTGNN | AGCRN | STSGCN | ASTGCN | STGCN | MTGNN | AGCRN | STSGCN |
| Time for Training Scratch | 73.183 | 20.564 | 13.913 | 28.235 | 37.899 | 270.531 | 60.341 | 30.905 | 46.031 | 127.651 |
| Time for Finetune | 57.232 | 17.886 | 14.620 | 17.894 | 34.167 | 243.172 | 52.114 | 40.865 | 30.303 | 115.901 |
| Time for Prompt Tuning | 50.818 | 13.675 | 9.327 | 12.013 | 24.733 | 216.587 | 37.187 | 17.220 | 20.125 | 70.851 |
| Faster x than Scratch | 30.560% | 33.500% | 32.962% | 57.454% | 34.740% | 19.940% | 38.372% | 44.281% | 56.280% | 44.496% |
| Faster x than Prompt | 11.207% | 23.543% | 36.204% | 32.866% | 27.611% | 10.933% | 28.643% | 57.861% | 33.587% | 38.869% |

Figure 5: Hyperparameter study on traffic prediction and crime prediction

utilize historical 12 time steps with 5 minutes a step to predict future 12 time steps on point-based datasets (PeMSD04, PeMSD08, PeMSD03, PeMSD07 and PeMS-Bay). And we use historical 6 time steps to predict future 1 time step on grid-based datasets (NYCTaxi, CHIBike and TDrive). All baseline methods follow their predefined settings as their papers.

## A.6 EFFICIENCY OF PROMPT TUNING ON CRIME PREDICTION AND TRAFFIC PREDICTION

To evaluate the model's ability to operate independently, we conducted efficiency experiments on traffic predictions using several state-of-the-art baselines. The results are presented in Table 9. From the results, we observed that our prompt tuning neural network significantly improved the efficiency of different baselines, reducing the time cost by approximately 10% to 57%. This finding further validates the advantage of the graph-based passing mechanism in terms of saving time. Additionally, we evaluated the efficiency of crime prediction, as shown in Figure 6. We compared our method's speed in crime prediction to the fine-tuning of a GNN-based model on the New York City and Chicago datasets. The results indicate that our method achieved a speed improvement of 23% to 28% compared to fine-tuning the GNN-based model on the New York City dataset. In the case of the Chicago dataset, our method outperformed fine-tuning by 3% to 10%. These findings highlight the advantage of our PromptST approach in real-life applications, particularly in the field of urban planning. Overall, results demonstrate that our PromptST framework offers improved efficiency across various tasks, making it highly suitable for real-life applications where efficiency is crucial.

## A.7 DESCRIPTION OF BASELINES

We compare 30 baselines including many state-of-art traffic flow prediction methods and crime prediction baselines, where are displayed as following:

- Traffic prediction methods. **DSANet** Huang et al. (2019): It is a method which adopts CNN for capturing temporal correlations and utilizes self-attention mechanism for capturing dynamic spatial information. **DCRNN** Li et al. (2018): To simulate spatial-temporal dependencies, a diffusion convolutional RNN with fusion process is used. **STGCN** Yu et al. (2018): To represent spatial-temporal coupling, it combines a gated temporal convolution module with a graph neural network.

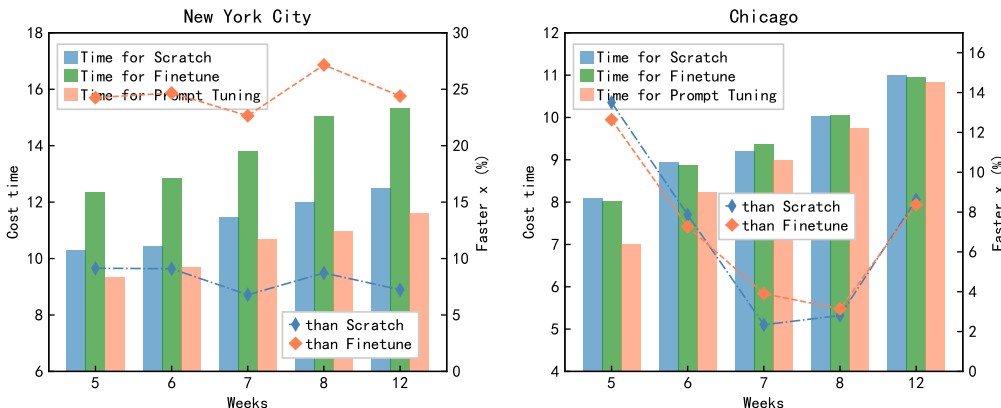

Figure 6: Time-consuming of crime prediction.

**GWN** Shleifer et al. (2019): It is a technique that combines 1D dilated convolutions and diffusion graph convolutions to capture spatial and temporal changes, enhancing the effectiveness of traffic prediction. **ASTGCN** Zhu et al.: It is an attention-based GCN model that additionally incorporates STGCN for capturing dynamic spatial and temporal information with spatial-temporal attention. **LSGCN** Han & Gong (2022): To capture spatial dynamics, it combines a graph attention network with a graph convolution network. And to capture temporal dynamics, it uses a temporal convolution network. **STSGCN** Song et al. (2020): By stacking numerous localized GCN layers with an adjacent matrix on the time dimension, it captures spatial-temporal correlations. **AGCRN** Bai et al. (2020): In order to capture spatial-temporal correlations, it uses learnt node embeddings in graph convolutions. **STFGNN** Li & Zhu (2021): The performance of traffic prediction is improved by using a spatial-temporal fusion graph neural network to capture spatial-temporal correlations. **STG-ODE** Fang et al. (2021): To address the limitiation caused by the neural networks' lack of depth, it uses differential equations. Shallow GNNs are unable to capture long-range spatial dynamics, and they ignore temporal dynamics, which are crucial for the task of traffic prediction. **Z-GCNETs** Chen et al. (2021a): For predicting traffic flow, it uses zigzag persistence along with a temporal-aware graph convolution network. **TAMP** Chen et al. (2021b): To capture dynamic spatial dependencies, it employs multiple persistence to collect temporal features, which are subsequently fed into graph convolutional networks. **DSTAGNN** Lan et al. (2022): The pre-defined static graph that is typically utilized in classic graph convolution is proposed to be replaced with a new dynamic spatial-temporal aware graph in this study that is based on a data-driven technique. Then, using an improved multi-head attention mechanism, it designs a novel graph neural network architecture that can not only represent dynamic spatial relevance between nodes but also acquire a wide range of dynamic temporal dependency from multiple receptive field features using multi-scale gated convolution. **FOGS** Rao et al. (2022): It is a technique that builds the association graph using the nodes' spatial-temporal dynamics. **STResNet** Zhang et al. (2017): It creates a complete STResNet structure based on the particular characteristics of spatial-temporal data. To describe the temporal closeness, period, and trend characteristics of crowd traffic, it specially uses the residual neural network framework. Based on data, STResNet learns to dynamically aggregate the output of the three residual neural networks. **DMVSTNet** Yao et al. (2018): To model both spatial and temporal relations, it suggests using a Deep Multi-View Spatial-Temporal Network (DMVSTNet) framework. This method specifically consists of three views: temporal, spatial, and semantic. The temporal view models correlations between future demand values with nearby time points using LSTM; the spatial view models local spatial correlation using local CNN. **STGNCDE** Choi et al. (2022): It explains how to use the STGNCDE method, which stands for spatio-temporal graph neural controlled differential equation. The concept of neural controlled differential equations (NCDEs) for processing sequential data is revolutionary. The idea is expanded, and two NCDEs are created: one for spatial processing and the other for temporal processing. **STTN** Xu et al. (2020): To increase the precision of long-term traffic flow forecasting, it suggests a unique paradigm of Spatial-Temporal Transformer Networks (STTNs) that concurrently use dynamical directed spatial dependencies and long-range temporal dependencies. **TFormer** Jin et al. (2023a): It suggests a brand-new model called Trafformer that combines temporal and spatial data into a single transformer-style model. In the spatial-temporal correlation matrix, TFformer enables each node at each timestamp to interact with each other node at each other timestamp in a single step. TFformer can detect intricate spatial-temporal relationships thanks to this design. **ASTGNN** Guo et al. (2021):

It creates a unique self-attention mechanism in the temporal dimension. In addition to enjoying global receptive fields that are advantageous for long-term forecast, it enables the prediction model to catch the temporal dynamics of traffic data. It creates a dynamic graph convolution module for the spatial dimension, using self-attention to capture the spatial correlations.

- Crime prediction methods. **STrans** Wu et al. (2020): By stacking two layers of Transformer to represent spatial-temporal links across spaces and time, it investigates the sparse crimes. For the aggregation of spatial and temporal information, self-attention with query/key transformations is used.. **DeepCrime** Huang et al. (2018): It is a representative baseline for crime prediction that first encodes the temporal embeddings of crime occurrences through time using a recurrent neural network. The next step is to further aggregate temporal representations with the attentional weights using the attention mechanism. **STDN** Yao et al.: A flow gating approach is introduced in this framework to capture the time-aware reliance between areas, and a periodic shifting attention is suggested to learn the temporal patterns between various time periods. **ST-MetaNet** Xu et al. (2018): This model is a meta-learning strategy that uses a GNN-based sequence-to-sequence paradigm to capture various spatial correlations and extract meta information relevant to a given location. **STSHN** Xia et al. (2022): This technique uses hypergraph connections between regions to carry out spatial message transfer between various geographic regions. A stationary approach is taken in building the region hypergraph. Two spatial path aggregation layers are chosen as the number. **DMSTGCN** Han et al. (2021): With the help of this method, the graph convolutional network is enhanced with dynamic and complex geographical and temporal data. The time-aware graph constructor is used to capture relationships between road segments.

