# OpenReview forum: "Simple Yet Effective Spatio-Temporal Prompt Learning"
_ICLR.cc/2024/Conference — ICLR 2024 Conference Withdrawn Submission_

### Official Review · Reviewer_gHhr · 2023-10-28

**Soundness:** 2 fair
**Presentation:** 1 poor
**Contribution:** 2 fair
**Rating:** 3
**Confidence:** 5

**Summary:**

This paper proposes a lightweight and effective prompt learning paradigm for spatio-temporal prediction. Basically, it uses the idea of prompt tuning from language model, to make the pretrained model adapting to downstream tasks. The authors claim that their framework, PromptST, can effectively generalize to unseen scenarios and adapt to varying spatio-temporal data distributions. Evaluated on traffic and crime prediction datasets, PromptST is shown to achieve state-of-the-art prediction accuracy while maintaining computational
efficiency.

**Strengths:**

1. Addressing distribution shift in spatial-temporal prediction is an important research problem, and therefore, this paper is well motivated.

2. The proposed prompt tuning approach is lightweight as compared to full parameter finetuning solutions, and show promising performance.

**Weaknesses:**

1. The comparisons are not fair, in that PromptST is compared with baselines without any pretraining. How promptST compared with other pretrained model? What is the # of parameters of PrompST, is the performance gain from a larger model?

2. In general, it is an incremental work. Pretraining-finetuning is a widely used paradigm in various areas, including spatial-temporal prediction. This paper uses prompt tuning for efficiency considerations, while prompt tuing from NLPs can be directly applied to promptST in this paper, making the approach less intriguing.

**Questions:**

1. It is not clear how the pretrained model is obtained in the experiments. For example, for traffic prediction, is the model pretrained on all traffic datasets used in the experiments?

2. How does the performance improve with the increasing of pretrained data?

3. How the prompt tuning appoach compared with other lightweight finetuning solutions, e.g., LORA?

4. What is "C" at page 5, after eqn 6?

---

### Official Review · Reviewer_Ajy6 · 2023-10-30

**Soundness:** 3 good
**Presentation:** 3 good
**Contribution:** 3 good
**Rating:** 5
**Confidence:** 5

**Summary:**

This paper proposes a lightweight and effective prompt learning paradigm named as ProptST for spatio-temporal prediction and crime detection. The proposed paradigm due with the dynamic spatio-temporal data distribution effectively and adaptively, which improves the effectiveness of spatio-temporal prediction. Extensive experiments show the performance of PromptST.

**Strengths:**

S1. Relevant problem of potential societal interest.
S2. Well written paper which is easy to follow.
S3. The proposed paradigm help the improvement of effectiveness and efficiency of spatio-temporal prediction and crime detection. The experimental evaluation is sufficient.

**Weaknesses:**

W1. There is little figure to explain the proposed paradigm, making the part of technologies incomprehensible (e.g., how to split data into pretrained, tuned and tested).
W2. It would be better if authors can provide a description or summary table of all datasets used. It is confusing about the dataset information such as dataset scale. Also, there is a dataset (Geolife:https://www.microsoft.com/en-us/download/details.aspx?id=52367) lost to be evaluated.
W3. Formatting errors exist, such as the title of Table 3 in Page 6.

**Questions:**

See the Weaknesses.

---

### Official Review · Reviewer_xmYg · 2023-10-30

**Soundness:** 2 fair
**Presentation:** 3 good
**Contribution:** 2 fair
**Rating:** 5
**Confidence:** 2

**Summary:**

This paper presents a prompt learning approach for spatiotemporal prediction tasks. It has been tested on data with distribution shift, adaptation and generalisation settings, and diverse benchmark datasets, including in traffic flow forecasting and urban crime prediction.
The major concern of this paper is the overclaim on the novelty and the experiment settings

**Strengths:**

The paper presents a laborious effort on prompt learning for several benchmark spatiotemporal prediction datasets.

**Weaknesses:**

The work presented in the current paper is not the typical widely recognized pretraining-prompting paradigm.

The main concern is that some statements are overclaimed. The paper claims adaptation and generalization. However, the setting of experiments is too simple for evaluating the generalization. The pretraining set, prompt tuning set, and the test set are from the same dataset. A better generalization evaluation should be something like pretraining on one dataset, then do the tuning and testing on another dataset. An even stronger setting is to test the adaptation ability across domains such as pretraining on traffic prediction and tuning/testing on the crime prediction.

Only showing two sensor nodes one-day’s data in case studies is not enough to demonstrate the adaptation/generalization. For example, the shown shift might be captured by daily patterns or weekly patterns. Hence, it is not convincing enough to support the claims.

**Questions:**

•	Why the performance of MTGNN is not reported in Table 1? According to Table 2, the best performance of PromptST is based on MTGNN backbone.

•	Need some clarification about the details of the experiment setting. For example, for the baselines, is the training data is [X_{t-T+1},
X_t] or X_{pre} (whether the X_{tun} is included in the training set of the baselines)?

•	How did you split X_{pre} and X_{tun}? From Section 4.3, the X_{tun} has three settings, but not sure what is the default one in Table 1 for example.

•	For Section 4.4, it describes 3) w/o Skip. However, in Table 6, it only has w/o data initial instead of w/o skip. What is w/o initial?

•	For case studies shown in Figure 4, which part is the data used to do prompt tuning?

---

### Official Review · Reviewer_5ref · 2023-11-01

**Soundness:** 3 good
**Presentation:** 3 good
**Contribution:** 3 good
**Rating:** 5
**Confidence:** 2

**Summary:**

In this paper, the author introduced a simple yet powerful spatio-temporal prompt learning paradigm aimed at enhancing the robustness and generalization ability of spatio-temporal prediction models in the presence of dynamic distribution shifts. The framework integrates a specially designed prompt neural network into pre-trained models, which involves generating informative spatio-temporal prompt context that captures the underlying patterns and dynamics in the downstream urban data. The PromptST has significantly improved the resilience of pre-trained models to distribution shifts and enhanced their adaptability to new data, and showed remarkable effectiveness across various spatio-temporal prediction tasks.

**Strengths:**

①	The experimental baseline selection is quite comprehensive.
②	The author performed extensive experiments on a range of spatio-temporal prediction tasks using diverse datasets to thoroughly evaluate the effectiveness, efficiency, and robustness of the framework, which can validate the superiority of the approach and enhance the reliability of the results.
③	The author has provided a web address for accessing the model implementation, which facilitate result reproducibility.
④	The experiments not only selected datasets from popular application domains within the domain but also included large-scale datasets.

**Weaknesses:**

①	The article does not provide a detailed explanation of Figure 1. In the methodology section, it primarily explains the model at a theoretical and formulaic level. Readers would understand it easier if there were a more detailed explanation to the structure of PromptST, even if it's simple.
②	In the ablation experiment, the author removed several components to demonstrate the necessity of each, but provided only data results without a detailed analysis and comparison.
③	The author provided a brief summary of the experimental results and concluded that PromptST performs better without delving into an in-depth analysis of the results. They also did not analyze the reasons for varying performance of PromptST under different backbones.
④	The article does not provide the limitations of PromptST, and it also lacks a detailed explanation of the application domains of PromptST.
⑤	The word “performence” is spelled incorrectly in Table 3.

**Questions:**

How do we apply the inspiration of the prompt-tuning techniques in the field of textual data to the PromptST model?
Comparing the data in Table 1 and Table 3, PromptST performs better in traffic prediction than in crime prediction. Why is that?
Comparing fine-tuning to training from scratch, why is the benefit of the pre-trained model uncertain?
Why is the impact of kernel size of TCN more significant for the crime prediction task? Are the reasons the same as those for the more pronounced impact of embedding dimensions?